# A Multi-Subsampling Self-Attention Network for Unmanned Aerial Vehicle-to-Ground Automatic Modulation Recognition System

**Yongjian Shen [1,\*], Hao Yuan [2], Pengyu Zhang [2], Yuheng Li [2], Minkang Cai [2] and Jingwen Li [1]**

[1]  Beijing University of Aeronautics and Astronautics, Beijing 100191, China; lijingwen@buaa.edu.cn

[2]  Beijing Research Institute of Telemetry, Beijing 100076, China; haoyuan_1@stu.xidian.edu.cn (H.Y.); pengpengdezahuodian@mail.nwpu.edu.cn (P.Z.); liyuheng@mail.nwpu.edu.cn (Y.L.); mkcai@stu.xidian.edu.cn (M.C.)

\*  Correspondence: shenyongshen@buaa.edu.cn

**Abstract:** In this paper, we investigate the deep learning applications of radio automatic modulation recognition (AMR) applications in unmanned aerial vehicle (UAV)-to-ground AMR systems. The integration of deep learning in a UAV-aided signal processing terminal can recognize the modulation mode without the provision of parameters. However, the layers used in current models have a small data processing range, and their low noise resistance is another disadvantage. Most importantly, large numbers of parameters and high amounts of computation will burden terminals in the system. We propose a multi-subsampling self-attention (MSSA) network for UAV-to-ground AMR systems, for which we devise a residual dilated module containing ordinary and dilated convolution to expand the data processing range, followed by a self-attention module to improve the classification, even in the presence of noise interference. We subsample the signals to reduce the number of parameters and amount of calculation. We also propose three model sizes, namely large, medium, and small, and the smaller the model, the more suitable it will be for UAV-to-ground AMR systems. We conduct ablation experiments with state-of-the-art and baseline models on the common AMR and radio machine learning (RML) 2018.01a datasets. The proposed method achieves the highest accuracy of 97.00% at a 30 dB signal-to-noise ratio (SNR). The weight file of the small MSSA is only 642 KB.

**Keywords:** automatic modulation recognition (AMR); deep learning; self-attention mechanism

## 1. Introduction

With the promotion of unmanned combat concepts, the enhancement of drone-mounting capabilities, and the increase in flight time, unmanned aerial vehicle-(UAV)-to-ground automatic modulation recognition (AMR) systems are increasingly used in modern, social, especially in emergency communications. UAVs carrying communication reconnaissance payloads can perform the reconnaissance of general radio stations, with strong concealment and long detection distances, and are widely used. The aim of communication reconnaissance is to analyze various parameters of intercepted radio signals and find suitable methods to demodulate them.

However, in the actual transmission process, a signal not only is affected by the antagonistic factors of the transmitter (such as the radio frequency chain), but also changes due to the type of interference and propagation environment, increasing the difficulty of the signal communication and analysis. To simplify the transmission process, AMR methods are proposed, which receive the modulated signal and recognize the modulation mode without the provision of parameters.

Signal modulation recognition can provide essential modulation information for the received radio signals, especially non-cooperative radio signals, which contain cognitive radio, spectrum sensing, signal surveillance, and interference identification [1–4]. Traditional

methods face difficulty when coping with the complex and growing types of transmitters. Given that AMR serves as a bridge between signal detection and demodulation, a simple and feasible approach that can be deployed on terminals outfitted on UAV platforms is sorely needed.

### 1.1. Related Work

#### 1.1.1. Traditional AMR Methods

Traditional AMR can be categorized as likelihood theory-based AMR (LB-AMR) [5,6] or feature-based AMR (FB-AMR) [7]. TB-AMR methods recognize modulation schemes by Bayesian estimation and have high computational complexity. FB-AMR methods analyze a large number of signals, extract interesting features, and determine the category of modulation methods using instantaneous time-domain [8], transform domain [9], and statistical [10,11] features. The AMR task is a regression problem with multiple dimensions. Traditional machine learning methods, such as decision tree [12] and support vector machine (SVM) [13], are easily realized. However, the performance of such methods is reduced when addressing complex or multiple-modulation schemes.

#### 1.1.2. DL-Based AMR Methods

FB-AMR must pre-train on a large set of signals and learn the features of each modulation scheme. Compared with LB-AMR, FB-AMR methods can approximate the ground truth with lower computational complexity and can fit various modulation schemes. Deep learning AMR (DL-AMR) methods are highly dependent on prior knowledge.

Since the first open access AMR dataset, Radio Machine Learning (RML) 2016.04C, was proposed, the convolutional neural network (CNN) has been introduced in the AMR task [14]. There are many methods based on deep learning, which can be divided into three groups, depending on their research content. Some researchers are concerned with enhancing the capability of models. Methods include long short-term memory (LSTM) [15], ResNet [16], the gated recurrent unit (GRU) [17], and deep learning blocks such as temporal attention [18]. LSTM and GRUs extract features by comparing information in the time dimension. They are more suitable for tasks with sequential data inputs, but a large amount of computation limits their performance in mobile terminals. Generally, to avoid overfitting, just two blocks are used in a model. Therefore, CNN-based networks [14,16] have been introduced in the AMR task. These models have a moderate number of parameters and moderate structure complexity for the same performance as LSTM and GRUs. However, the current models all have a small processing range, and the signal data are too long for the models to process efficiently.

A second group of researchers have attempted to increase the channel's number of inputs to improve the classification accuracy. Normally, the inputs of models are signals with two channels, namely in-phase ($I$) and quadrature-phase ($Q$) channels, which are supported in public datasets. Researchers use signal processing methods to expand the number of input channels. The amplitude and phase, which can be easily calculated from $I$ and $Q$, are frequently used. Networks that expand the channels of inputs [19–21] greatly increase the computational cost, and additional factors that increase the requirement of inputs betray the initial goal of easily realizing the AMR task. In preprocessing inputs before these are input to the models, some methods are too specialized and complex to be applied to AMR tasks on a large scale. The models have more parameters, with a deeper and wider network structure.

The last improvement direction of the AMR task is converting the signal inputs into images. The most representative methods use three-channel constellation images as inputs [22,23] and use AlexNet [24] and GoogLeNet [25] for image classification. However, these methods make the AMR task more complex because the input is usually a signal rather than an image.

Comprehensively considering the advantages and disadvantages of the three groups of methods, this paper follows the idea of the first group. These methods have a small

processing range in each layer but have weaknesses in terms of noise resistance. However, noise in signal transmission is unavoidable. Moreover, even with LSTM and recurrent neural networks (RNNs), which are common in the analysis of sequence data, either the computation or the number of parameters is too much for deployment on mobile terminals mounted on UAVs, which means that they are unsuitable for UAV-to-ground AMR systems.

### 1.2. Contributions

The performance of uniform subsampling is equal to that of no subsampling when the subsampling rate is 2 [26]. By reducing the input signal length by subsampling, the parameters and computation in models are greatly decreased. We propose a multi-subsampling self-attention network (MSSA) that can be deployed on a terminal in our UAV-to-ground AMR system. Our main contributions are summarized as follows:

- We design an information integration module with ordinary convolution and dilated convolution branches. Dilated convolution has a larger receptive field than ordinary convolution and is more suitable for global information extraction. The sum of the two branches provides more detailed information.
- To enhance the noise resistance, we introduce a self-attention module with a strong feature extraction capability. The module can dynamically adjust the weights of parameters to amplify the influence of those that are beneficial for modulation recognition and diminish the influence of invalid parameters during the recognition process.
- We subsample the signal into multiple signals with two branches, $I$ and $Q$, and concatenate them channel-wise. We finesse the model architecture to prevent overfitting. We propose MSSAs in large, medium, and small sizes, with fewer parameters and faster speeds, which are more suitable for our UAV-to-ground AMR system.
- Ablation experiments on a common dataset with current models show the ability of the proposed method in AMR. MSSA has the best performance on RML 2018.01a and 97.00% accuracy when the signal-to-noise ratio (SNR) is 30 dB. Different sizes of MSSA each have their advantages in terms of accuracy, speed, and parameters. The weight file of MSSA(S) is only 652 KB.

### 1.3. Organization

The remainder of this paper is organized as follows. Section 2 presents the system model. Section 3 analyzes the structure and theory of the proposed method. Section 4 discusses experiments on the dataset, including the comparison of current models and different hyperparameters. Section 5 provides our conclusions.

## 2. System Model

The emergence of unmanned aerial vehicles (UAVs) has revolutionized the field of remote sensing and other aerial applications by providing both low-altitude and high-altitude platforms for data collection [27]. Depending on the payloads mounted, a UAV can serve as either a computational server or a relay [28], thus enabling diverse ranges of applications in mapping, wildlife conservation, and emergency communications [29], among others.

Our UAV-to-ground AMR system consists of a reconnaissance drone and a ground control terminal, as shown in Figure 1. The drone conducts reconnaissance on communication links such as air-to-air, air-to-ground, and ground-to-ground. The reconnaissance equipment includes various types of airborne radios, radars, vehicle radios, and handheld radios. The ground control terminal, functioning as either a high-capacity computer or server cluster, can receive brief reconnaissance results in real-time and analyze reconnaissance data after the drone returns. AMR methods are typically implemented on the ground control terminal due to the heavy computational burden involved. The information fed back by the drone includes the number of reconnaissance signals, as well as the information such as the direction, time, frequency, modulation method, power, and possible transmission source type of each signal. The processing of reconnaissance data by ground control

terminals can achieve signal demodulation and analysis. Nevertheless, due to the low computational cost of our proposed AMR methods, the UAV-mounted payloads with embedded microprocessors can also perform signal demodulation and analysis. The ground terminal solely performs signal behavior analysis and transmits relevant instructions to the drone.

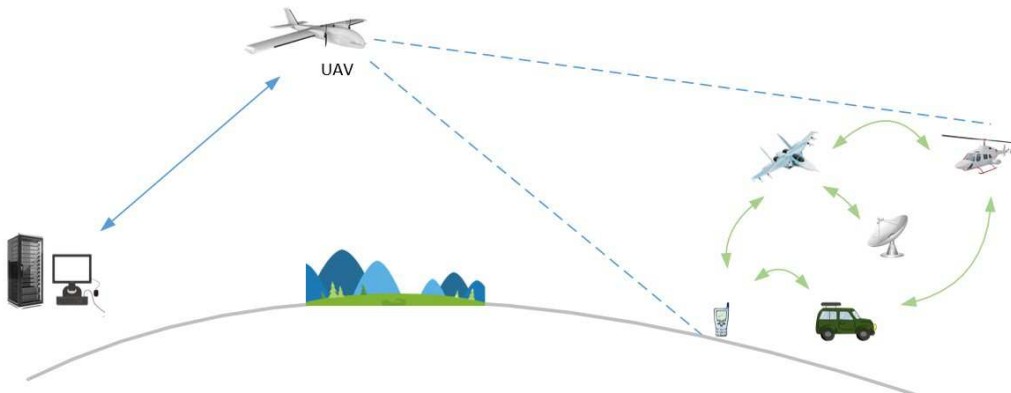

**Figure 1.** Schematic diagram of a UAV-to-ground AMR system.

Our system can also be utilized as a UAV-assisted mobile edge computing (MEC) architecture, in which the UAV and ground control terminal jointly demodulate and analyze the received signal before transmitting it to end users. Additionally, a flying ad hoc network (FANET) comprising multiple UAVs could be integrated into our system to reduce response delays and increase response probabilities [30]. By receiving signals from various transmitter sources, the UAVs can also support signal modulation analysis, significantly contributing to emergency communications.

## 3. Design and Implementation of Multi-Subsampling Self-Attention Network

### 3.1. Architecture

Although current DL-AMR methods have high accuracy, the complexity is high. Except for expanding the number of input channels by preprocessing, many complex modules are used in DL-AMR. The optional expanding channel can obtain the amplitude and phase through the provided $I/Q$ signal [15], and the models used for DL-AMR include LSTM, DAE [31], and GRU [32]. The amplitude $A$ and phase $\phi$, which can be easily calculated from $I$ and $Q$, with the formula:

$$x = \left[ \begin{array}{c} A \\ \phi \end{array} \right] = \left[ \begin{array}{c} \sqrt{I^2 + Q^2} \\ \arctan 2(Q/I) \end{array} \right]. \tag{1}$$

However, the received signal in common applications is typically represented as $I$ and $Q$ components, which requires specialized knowledge to process the expanded channels, including the amplitude, phase [19–21], and constellation image [22,23], as used in these modulation schemes. In recent years, the structure of deep learning has become deeper and more complex, with higher computational costs. For signal data, the parameters can be limited using simple CNN networks and are sufficient for the modulation classification task. We selected a CNN as our main framework.

Our network aims to achieve a more balanced architecture that minimizes the computational cost, accelerates the training speed, and requires less prior knowledge. Our experiments show that the original input with a length of 1024 is excessively long for this task, which can be attributed to the large number of parameters in the models used. Thus, before the CNN module, we reshape the $(1024, 2)$ inputs into $(4, 256, 2)$ or $(2, 512, 2)$ to reduce the length, through uniform subsampling with two subsampling rates. This expands the channels of inputs without changing the original data, and the decrease in input length reduces the number of parameters. It will greatly reduce the inference speed

when embedded on the terminal. We chose to utilize multiple cascaded residual modules following the first layer of our network. The residual mechanism allows for the preservation of original information during the feature transmission process. Then, we use dilated convolution, which can extract features from greater global information, and an ordinary convolution layer for local information.

With extracted local and global information, the addition of two branches facilitates the subsequent self-attention task. The self-attention mechanism has a strong capability to extract interesting information, which will affect the results of category classification. The traditional fully connected layer generally has more parameters than the convolution layer and benefits convergence. We put the fully connected layer at the two last layers to obtain the features of the signal, and a softmax function distributes the probabilities of each category. The number of dense units should match the number of modulation schemes used. By utilizing subsampled inputs, we can significantly reduce the number of trainable parameters in the last two layers of our model, which tend to be the main source of computational cost. Figure 2 shows the main framework of our model with a subsampling rate of 4. Figure 3 depicts the primary architecture of ResNet as presented in [16]. Our method employs fewer kernels in each convolutional layer compared to ResNet. Additionally, the input size of the first fully connected layer in ResNet is $(32 \times 16 \times 1)$, while our model's input size is $(16 \times 8 \times 1)$. Consequently, the number of trainable parameters in our approach has been reduced by a factor of four.

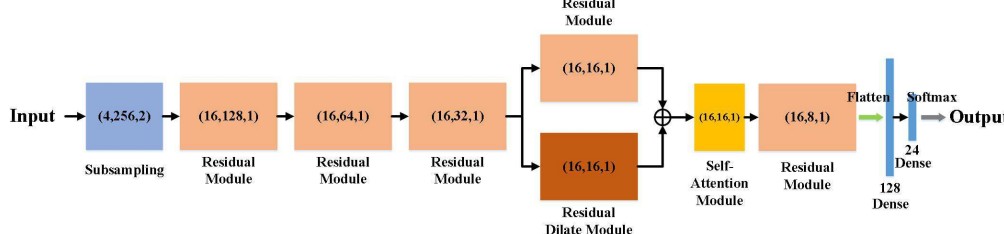

**Figure 2.** Structure of MSSA(M) when the subsampling rate is 4. The number of last dense units is the same as the number of types of modulation schemes.

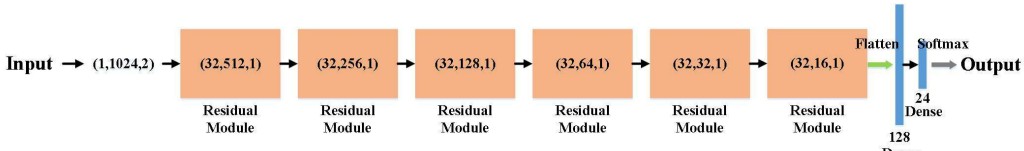

**Figure 3.** Structure of ResNet [16]. Our method has fewer kernels in each convolutional layer.

We selected suitable hyperparameters to fine-tune the model, including the number of convolution layers, kernel size, learning rate, and the sample size of each batch. The key point in DL-AMR is to solve the overfitting of models. The latest model, especially one with a self-attention module, could easily be subject to overfitting because of the higher computation cost. We conducted experiments on different hyperparameters and selected the best.

### 3.2. Methodology

#### 3.2.1. Enhanced Processing Range via Dilated Residual Connections

As the signal has fewer dimensions, the length of the signal data is greater than that of the image data. The receptive field is defined as the region in the input space that a particular CNN feature is looking at. The input of the common AMR dataset has a length of 1024. However, ordinary convolution always has fewer receptive fields, which are 3 with a 3 kernel size, and successive convolution layers have 5 receptive fields. Let $k$ be the

kernel size, and $s$ be the stride of the convolution layer. Then, the receptive field of the convolution layer $CF$ can be calculated as:

$$CF_i = F_{i-1} + ((k-1) * \prod_{j=1}^{i-1} s_j), \tag{2}$$

where $CF_i$ refers to the receptive field of the current convolution layer, and $F_{i-1}$ is the previous level. Hence, we serially connect four convolution layers as the base module. The stride and kernel sizes are set to 1 and 3, respectively. The final receptive field is 9, which cannot cover the entire length of the signal. The number of convolution layers in the residual modules needs to be controlled to avoid overfitting, but this number has already reached the limit of the modules.

Unlike ordinary convolution, dilated convolution conducts a convolution operation whose kernel has holes. Figure 4 shows the different kernels of two convolutions. By filling kernels with holes, the receptive field of dilated convolution $DCF$ is increased,

$$DCF_i = F_{i-1} + ((k * r - 1) * \prod_{j=1}^{i-1} s_j), \tag{3}$$

where $r$ is the dilated rate of dilated convolution. Then, the whole receptive field of the residual dilated module is 11 when $r = 2$ and $k = 3$. The expanded receptive field provides more global information. In contrast to the pooling layer, the dilated convolution layer does not remove elements with smaller values, which can retain the most information of inputs. Dilated convolution operates on data at equally spaced intervals, effectively performing a form of specialized subsampling. Consequently, the features extracted from dilated convolution will differ from the features extracted by parallel residual modules. In addition, to maintain gradient stability, we set a bridge between every two layers. The structure is shown in Figure 5. The residual block output is:

$$H_{Res} = x + W_i \times (W_{i-1} \times x + b_{i-1}) + b_i, \tag{4}$$

where $W \times x + b$ represents convolution, and $x$ is the input of the one residual block. The gradient of this module is:

$$\begin{aligned} \frac{\partial H_{Res}}{\partial x} &= 1 + \frac{\partial W_i \times (W_{i-1} \times x + b_{i-1}) + b_i}{\partial x} \\ &= 1 + W_i \times W_{i-1}, \end{aligned} \tag{5}$$

where 1 keeps the gradient in a controllable range. This can avoid the explosion and disappearance of the gradient. We combine one convolution with a $(1, 1)$ kernel size, two residual blocks, and one max-pooling layer. The $(1, 1)$ convolution layer extracts the channel-wise information and the $(3, 1)$ layer extracts the height and channel dimensions. The $(2, 1)$ max-pooling layer reduces the width of the inputs. Due to the $(channel, length, 2)$ input shape, the kernel size in the first residual module is set as $(3, 2)$ to consider the information in both $I/Q$ signals. Similarly, the kernel size of the max pooling layer in the first residual module is set as $(2, 2)$ to reduce the dimension in width. We replace the $(3, 1)$ convolution layer with a $(3, 1)$ dilated convolution layer in the residual dilated module.

The larger receptive field of residual dilated convolution can cover information that ordinary convolution cannot, and the features extracted by dilated convolution can be regarded as information from larger-size inputs. Ordinary convolution and dilated convolution branches in the proposed model provide features extracted from different scales to the next self-attention module.

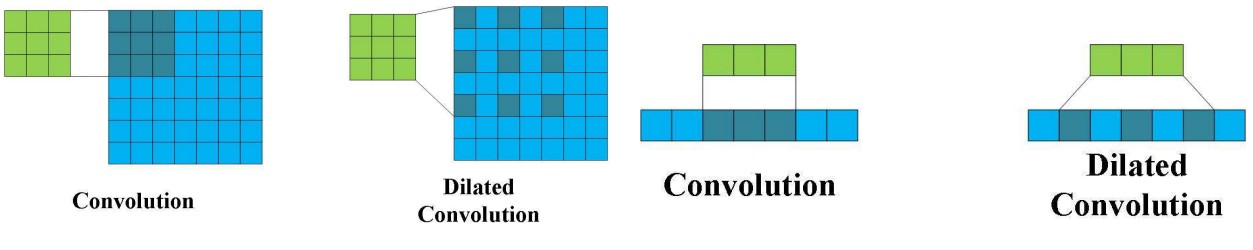

**Figure 4.** Two types of convolutions with data of different dimensions. By filling kernels with holes, the receptive field of dilated convolution is increased.

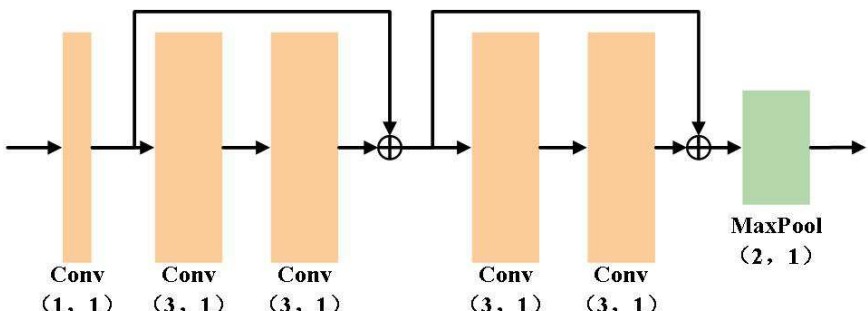

**Figure 5.** Structure of the residual module. Each block has two convolution layers with a kernel size of $(3, 1)$ and one max-pooling layer with a kernel size of $(2, 1)$; however, the sizes of these layers are $(3, 2)$ and $(2, 2)$, respectively, when in the first residual module of the MSSA.

### 3.2.2. Enhanced Robustness of Attention Models against Noise

After the two branches of convolution and dilated convolution, a module that can integrate global and local information to extract interesting features is needed. Therefore, a self-attention module is introduced. The attention model can filter ineffective information [33], which can be noise generated in the transmission process or information repeated in a periodic signal. The output of attention module $H_A$ can be formulated as:

$$H_A = \sigma(Mask(x)) \odot \phi(Trunk(x)), \tag{6}$$

where $Mask(x)$ represents the gate controlled by the output of $Trunk(x)$. $Mask(x)$ and $Trunk(x)$ can be any type of structure and are the outputs of the mask and trunk branch. The mechanism assumes the weights for elements in each location, and the trainable weights can dynamically extract interesting features to enhance the classification ability of models.

Comparing AMR and image classification, the AMR task is relatively simple. The number of parameters and layers should be controlled to avoid wasting computational resources. We use the attention mechanism once in the proposed model. We adjust the structure of the attention module, i.e., the self-attention module [34],

$$
\begin{aligned}
H_{SA} &= P_{QK}(x) \times V(x) \\
&= softmax\left(\frac{Q(x) \times K(x)}{\sqrt{d_K}}\right) \times V(x),
\end{aligned}
\tag{7}
$$

where $Q(x), K(x)$, and $Vx$ refer to the extracted features in three branches, and $d_K$ is the number of channels. In this module, $Q(x) \times K(x)$ strengthens the interesting elements, and the softmax function assumes the probabilities of elements in each location. Then, the weights $P_{QK}(x)$ generated by $softmax\left(\frac{Q(x) \times K(x)}{\sqrt{d_K}}\right)$ select the features in $V(x)$ as the output of the self-attention module. In the calculation of DL-AMR, the features can be a matrix with shape $(batch, length, 1, channels)$, with a dimension of 1. Then, the result of $P_{QK}(x)$ in

$H_{SA}$ would have the shape $(batch, 1, 1, channels)$. The probabilities for $V(x)$ are constant values that are inefficient for the classification task. Therefore, we use the element-wise product $\odot$ to replace matrix multiplication $\times$, as in Equation (6), with the formula:

$$
\begin{aligned}
H'_{SA} &= x + Relu(P'_{QK}(x) \odot V(x)) \\
&= x + Relu(softmax(Q(x) \odot K(x)) \odot V(x)).
\end{aligned}
\tag{8}
$$

The kernel size in $Q$ is $(1,1)$, and the others are $(3,1)$. The $Q$ branch learns the characteristics channel-wise. Then, the impact factors in the channel and location are $Q(x) \odot K(x)$. The probabilities can be obtained by the softmax function. Activated by the ReLU function, $Relu(P'_{QK}(x) \odot V(x))$ contributes to extracting key features in both local and global information. According to the gradient formula of parameter $W_V$ in $V(x)$,

$$
\begin{aligned}
\frac{\partial H'_{SA}(x')}{\partial W_V} &= Relu'(P'_{QK}(x') \odot V(x')) \frac{\partial P'_{QK}(x') \odot V(x')}{\partial W_V} \\
&= Relu'(P'_{QK}(x') \odot V(x')) P'_{QK}(x') \odot \frac{\partial V(x')}{\partial W_V} \\
&= Relu'(P'_{QK}(x') \odot V(x')) P'_{QK}(x') \odot x',
\end{aligned}
\tag{9}
$$

where $x'$, $P'_{QK}(x')$, and $Relu'(P'_{QK}(x') \odot V(x'))$ are constants in the parameter update process, which could be controlled by $Q(x)$ and $K(x)$. Similarly, the parameters in $Q$, $K$, and $V$ are all learnable and trainable, and they can influence each other. Following extensive training, the value of these valid parameters can be adjusted, resulting in a significant amplification of their impact on modulation recognition. Additionally, dynamic parameters can filter out noisy data and place greater emphasis on valid data. The information from the addition of the convolution and dilated convolution layers is effectively utilized. Like the residual module, the addition operation is used before the output of the self-attention module.

As a result, the gradient of this module becomes controllable and the noise resistance is increased. It avoids overfitting because of the addition with input $x$. The structure of our self-attention module is shown in Figure 6. Ablation experiments demonstrate the capability of the self-attention mechanism in the AMR task. However, self-attention increases computation, which could affects the deployment on a mobile terminal.

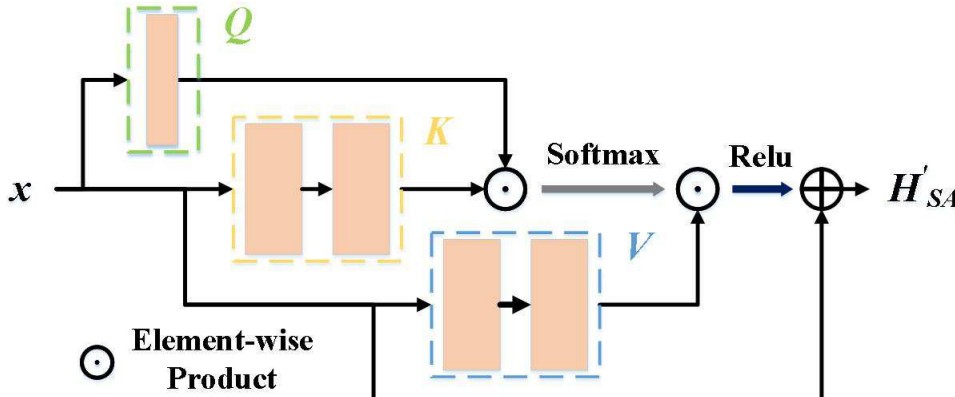

**Figure 6.** Self-attention module. Kernel size in $Q$ is $(1,1)$; others are $(3,1)$. We add input $x$ and output $H'_{SA}$ to prevent the gradient exploding and disappearing.

### 3.2.3. Streamlined Modeling with Subsampling Layer

To simplify the models and decrease the number of parameters, a subsampling layer is added at the start of our network. Ramjee et al. conducted experiments on uniform, random, and magnitude rank subsampling [26]. Uniform subsampling had an equal

performance with no subsampling when the subsampling rate was 2, while the others had no improvement.

Hence, we subsample the data with a constant interval and concatenate them channel-wise. Then, the length of inputs will be reduced, and the number of channels increased. However, the original network is too complex to learn the features in shorter data, which will cause overfitting in the training process. Therefore, the hyperparameters in the other three sizes of MSSA are adjusted to be suitable for shorter data after subsampling, which include the number of residual modules, residual module filters, and fully connected layers (dense layers) units. This greatly reduces the number of trainable parameters and increases the training speed. Table 1 shows the details of each model size.

**Table 1.** MSSA with different hyperparameters.

| Hyperparameters | Inputs Shape | Residual Modules | Residual Module Filters | Dense Units |
|---|---|---|---|---|
| **MSSA(XL)** | (1, 1024, 2) | 6 | 32 | 128 |
| **MSSA(L)** | (2, 512, 2) | 6 | 32 | 128 |
| **MSSA(M)** | (4, 256, 2) | 5 | 16 | 128 |
| **MSSA(S)** | (4, 256, 2) | 5 | 16 | 64 |

Finally, we proposed four different models featuring two subsampling rates. The utilization of dilated convolution provided an additional specialized subsampling result for each model. We compared our methods with current models on a common public dataset, showing good performance. The proposed method has a 60.90% mean classification accuracy on SNR from −20 dB to 30 dB. Compared with current models, MSSA(S) has the fewest parameters and the fastest training speed, but slightly less accuracy. It is more suitable for signal detection and recognition systems.

### 3.3. Equipment and Facilities

Ablation experiments were conducted on a 64-bit Linux system equipped with an Nvidia GeForce RTX 2080 Ti graphics card with 12 GB memory. All models were trained with TensorFlow v1.14, CUDNN v7.4, and CUDA v10.0. We used the Adam optimizer with a learning rate of $1 \times e^{-4}$. The batch size was 1000 and iterations were limited to 100, except that the iterations of the other three MSSAs were 200. We saved the best model at each epoch. The loss function used the category cross-entropy function.

Figure 7 shows the deployment details of our UAV-to-ground AMR system. We selected an Nvidia Jetson TX2 as the signal processing terminal and a MicroPhase ANTSDR E310 transmitter. We leveraged TensorRT on the terminal to accelerate the inference process of (MSSA(S)).

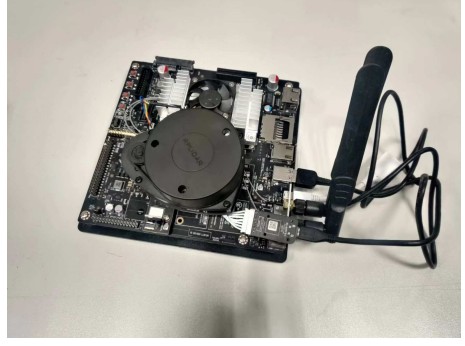
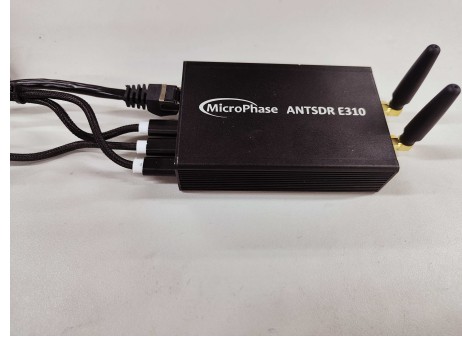

(**a**) Signal processing terminal        (**b**) Transmitter

**Figure 7.** Configuration of Nvidia Jetson TX2 signal processing terminal and MicroPhase ANTSDR E310 transmitter.

## 4. Experiments and Results

For evaluation, we selected the RML 2018.01a dataset [16], which is a common AMR dataset. This dataset contains 24 types of modulation schemes: OOK, 4ASK, 8ASK, BPSK, QPSK, 8PSK, 16PSK, 32PSK, 16APSK, 32APSK, 64APSK, 128APSK, 16QAM, 32QAM, 64QAM, 128QAM, 256QAM, AM-SSB-WC, AM-SSB-SC, AM-DSB-WC, AM-DSB-SC, FM, GMASK, and OQPSK. There were 2555904 sample signals with a shape of $(2, 1024)$. The signal-to-noise ratio (SNR) of signals in the dataset ranged from $-20$ dB to 30 dB with an increment of 2. Each modulation scheme had 26 sets of signals with different SNRs, and each SNR set had 4096 signals. Compared with other public AMR datasets, this dataset has more samples and more kinds of modulation schemes. Table 2 compares the properties of the main AMR datasets. The signals have two branches, *I* and *Q*. We trained the models with all kinds of modulation schemes, conducted ablation experiments on each hyperparameter, and proposed the best couple of hyperparameters. The dataset was divided into training and test sets at a 7:3 ratio. There were 1,789,128 signals in the training set and $766, 776$ in the test set.

**Table 2.** Comparison of open AMR datasets.

| Dataset | Number of Modulation Schemes | Sample Dimension | Dataset Size | SNR Range (dB) |
|---|---|---|---|---|
| **RML 2016.04c** | 11 | $2 \times 128$ | 162,060 | $-20$:2:18 |
| **RML 2016.10a** | 11 | $2 \times 128$ | 220,000 | $-20$:2:18 |
| **RML 2016.10b** | 10 | $2 \times 128$ | 1,200,000 | $-20$:2:18 |
| **RML 2018.01a** | 24 | $2 \times 1024$ | 2,555,904 | $-20$:2:30 |

LSTM had the best performance when tested on the RML 2018.01a dataset and was the main comparison network [1]. In addition, MSSA is an improvement network based on ResNet, and a CNN is the baseline of the AMR task. It is necessary to compare MSSA with CNN and ResNet.

Our experiment had two main purposes, the more important one being to demonstrate the modulation recognition ability of the proposed model compared with current or basic models, and the other being to provide a faster and simpler network while retaining most of the performance. We display the results of comparisons with other models in Section 4.1, and Section 4.2 shows the improvement of our model in terms of adjusting the structure and hyperparameters.

### 4.1. Experimental Comparison for AMR Task

To fairly test the ability of models in radio modulation recognition, no data enhancement methods were used, not even random shuffling, because this would affect the gradient updates when the number of samples was too large. The inputs in this experiment were with the shape $(1024, 2)$, which was reshaped to a $(1, 1024, 2)$ tensor by reshaping the layer with the data format "channels first". The network was fed one of the types of modulation schemes and estimated the probability of the 24 classes. The prediction of a model is the greatest value in the confidence matrix. Figure 8 shows the recognition accuracy curves of each class of models in each SNR. The proposed model had a higher accuracy in most SNRs, except SNRs in the range of $-5 \sim 8$ dB. The CNN in [14] performed the worst among the four models.

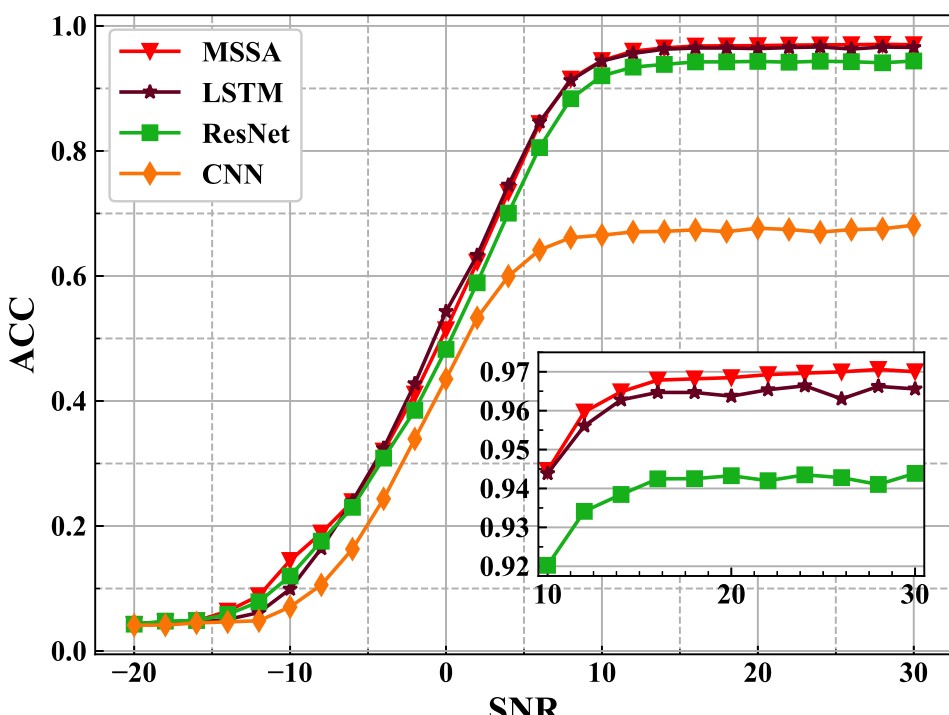

**Figure 8.** Recognition accuracy of models on the RML 2018.01a dataset. The proposed MSSA has the highest accuracy when SNR is above 10.

The other evaluation index of AMR is the confidence confusion matrix of models, which will reflect the quality and performance of the classifier. It demonstrates the certainty and correctness of the classifier's classification results for each category during the prediction phase. By observing the confidence confusion matrix, we can identify which categories have more accurate classification results, which categories are often confused, and the performance of the classifier when classifying samples with high uncertainty. We tested samples with different SNRs and calculated the mean confidence indexes of each class when inputting a signal. Then, confusion results are visualized in Figure 9. The vertical axis on each matrix denotes the true labels, and the horizontal axis denotes the predicted labels. As shown in Figure 8, MSSA performed the best in the confidence confusion matrix. MSSA had one modulation scheme, 16QAM, which was difficult to recognize, while LSTM and ResNet had two modulation schemes.

For the SNR of signals that are almost above 6 dB or 8 dB in the realistic AMR task, this paper shows the accuracy of models when the SNR is 6, 14, 22, and 30 dB. We calculated the average classification recognition accuracy without distinguishing the SNR and classification. The mean class accuracy values of models are shown in Table 3. The proposed model had the highest accuracy when the SNR was above 14 dB, and it had the highest mean accuracy. From ResNet to MSSA, the average accuracy increased by 2.09%. This experiment shows the capability of our models in the AMR task.

**Table 3.** Comparison of models in terms of classification recognition accuracy (%).

| Acc (%) in SNR (dB) | 6 | 14 | 22 | 30 | Mean (−20:2:30) |
|---|---|---|---|---|---|
| CNN | 64.16 | 67.14 | 67.43 | 68.12 | 43.89 |
| ResNet | 80.54 | 93.85 | 94.20 | 94.39 | 58.81 |
| LSTM | **84.65** | 96.28 | 96.54 | 96.59 | 60.22 |
| MSSA | 84.38 | **96.49** | **96.93** | **97.00** | **60.90** |

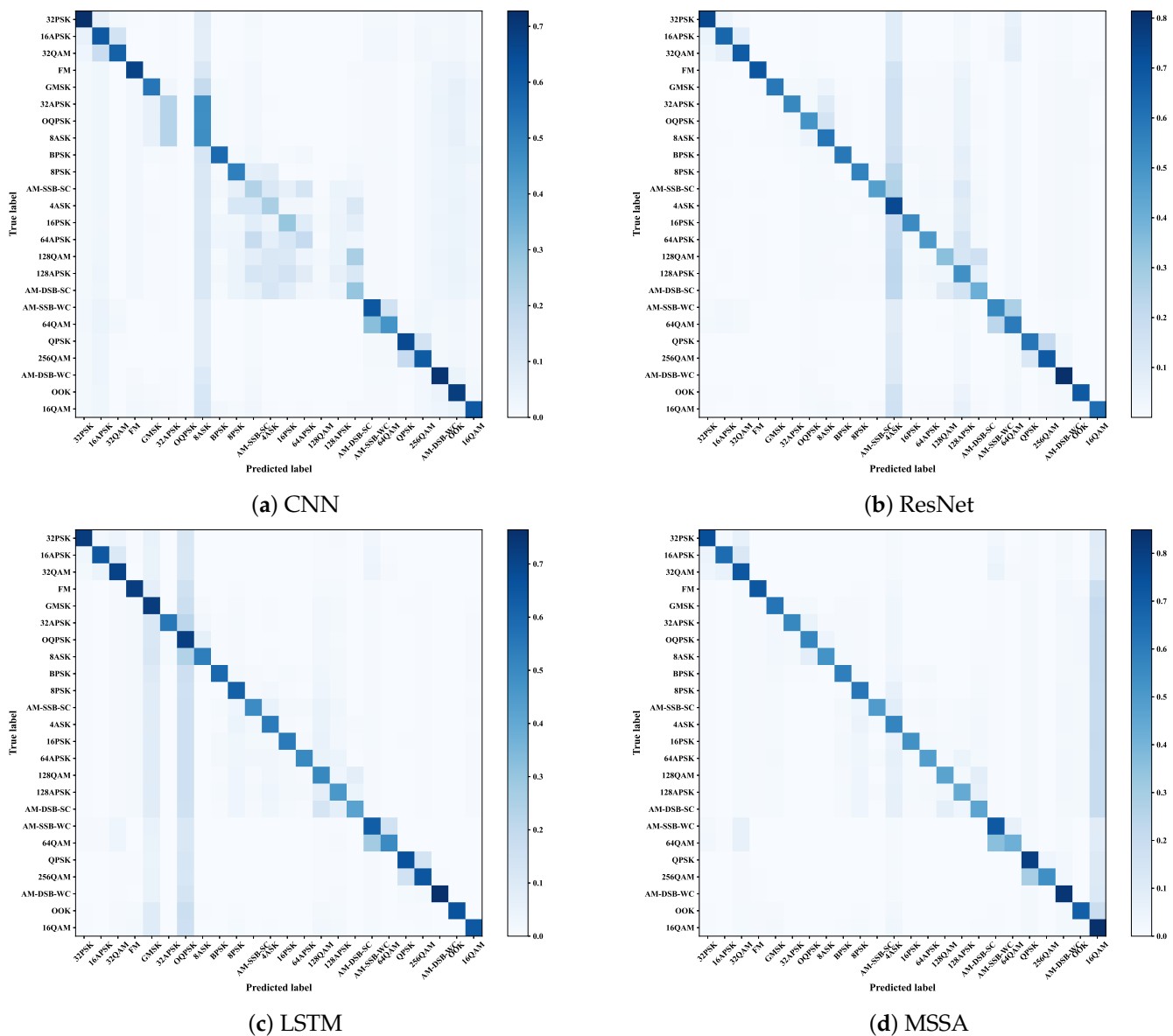

**Figure 9.** Confidence confusion matrices of the models. MSSA has one modulation scheme, 16QAM, that is difficult to recognize.

### 4.2. Experimental Comparison on Hyperparameters

This paper adjusted the structure of the model with a subsampling rate of 4, as shown in Table 1, because the length of the data is too short to achieve a higher mean classification accuracy when trained on the original model. The accuracy values of models are shown in Table 4, where the indicator values are shown for the convenient comparison of the training speed, parameters, and classification recognition accuracy. The number of trainable parameters in a model is strongly correlated with the utilization of computational and memory resources. Increasing the number of parameters may lead to a higher computational load on the terminal or embedded microprocessor, thus reducing both the training and inference speeds. MSSA(L) and MSSA(S) had the fastest speeds. MSSA(S) used the fewest parameters, while the accuracy of MSSA(L) was higher. While MSSA(L) had more parameters, it had the same training speed as ResNet and higher accuracy. Figure 10 shows the accuracy curves of models in each SNR.

**Table 4.** Comparison of models on size, complexity, and classification recognition accuracy (%).

| | Time (Second/Epoch) | Parameters | SNR = 6 (dB) Acc (%) | SNR = 30 (dB) Acc (%) | Mean Acc (%) |
|---|---|---|---|---|---|
| **CNN** | 367 | 13,064,524 | 64.16 | 68.12 | 43.89 |
| **ResNet** | 171 | 139,192 | 80.54 | 94.39 | 58.81 |
| **LSTM** | 1242 | 202,766 | **84.65** | 96.59 | 60.22 |
| **MSSA(XL)** | 283 | 218,200 | 84.38 | **97.00** | **60.90** |
| **MSSA(L)** | 171 | 152,696 | 82.09 | 95.78 | 59.70 |
| **MSSA(M)** | **99** | 54,632 | 75.80 | 93.48 | 57.01 |
| **MSSA(S)** | **99** | **36,648** | 72.43 | 90.50 | 55.25 |

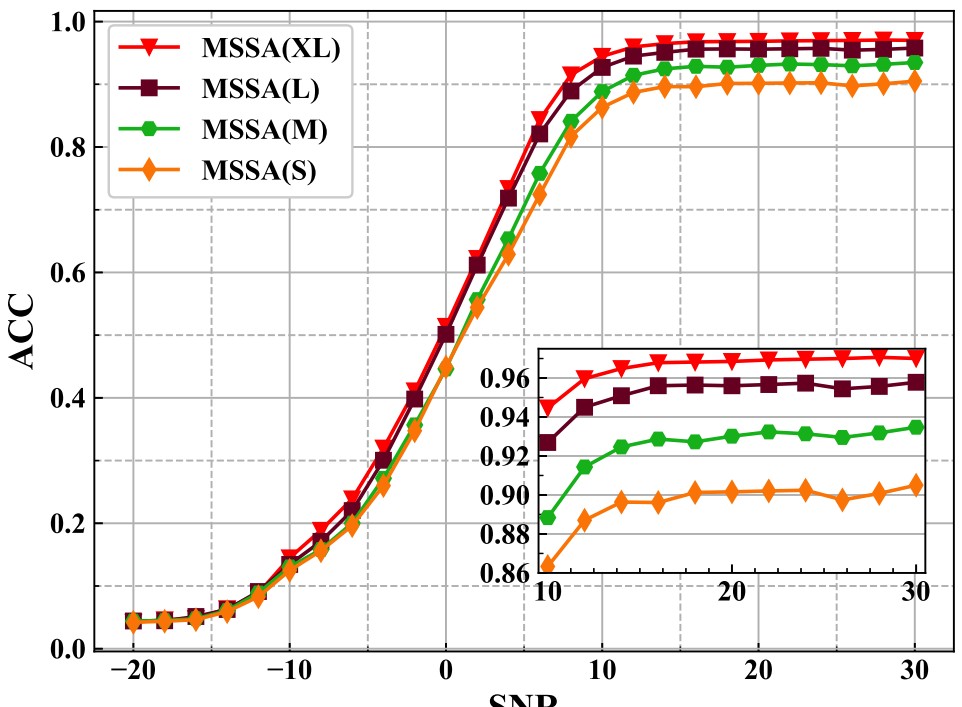

**Figure 10.** Recognition accuracy of MSSA with different hyperparameters on the RML 2018.01a dataset. As the subsampling rate increases, the model has fewer parameters, and the accuracy decreases by approximately 2.5%.

The capabilities of models decreased as the number of parameters decreased, but the speeds increased. These models with different sizes all outperformed the CNN, and even at the same training speed, MSSA(L) was better than ResNet. These four MSSAs in different sizes could be applied to different situations with different requirements, such as lower parameters or faster speeds. Figure 11 shows the confusion matrices. The OOK modulation scheme was difficult for MSSA in the M or S size. Although they have a problem recognizing signals with OOK modulation, the confidence indexes of the OOK scheme are high (above 0.8).

We proposed three sizes of MSSA models, with faster speeds and fewer parameters. They have their own advantages. MSSA(XL) performed the best among current models, with a few more parameters and a slower training speed. The L-sized MSSA(L) was the most recommended model, with a moderate performance in terms of accuracy, training speed, and parameters. MSSA(M) was more suitable when speed was required. MSSA(S) was more suitable when the parameter requirement was very strict.

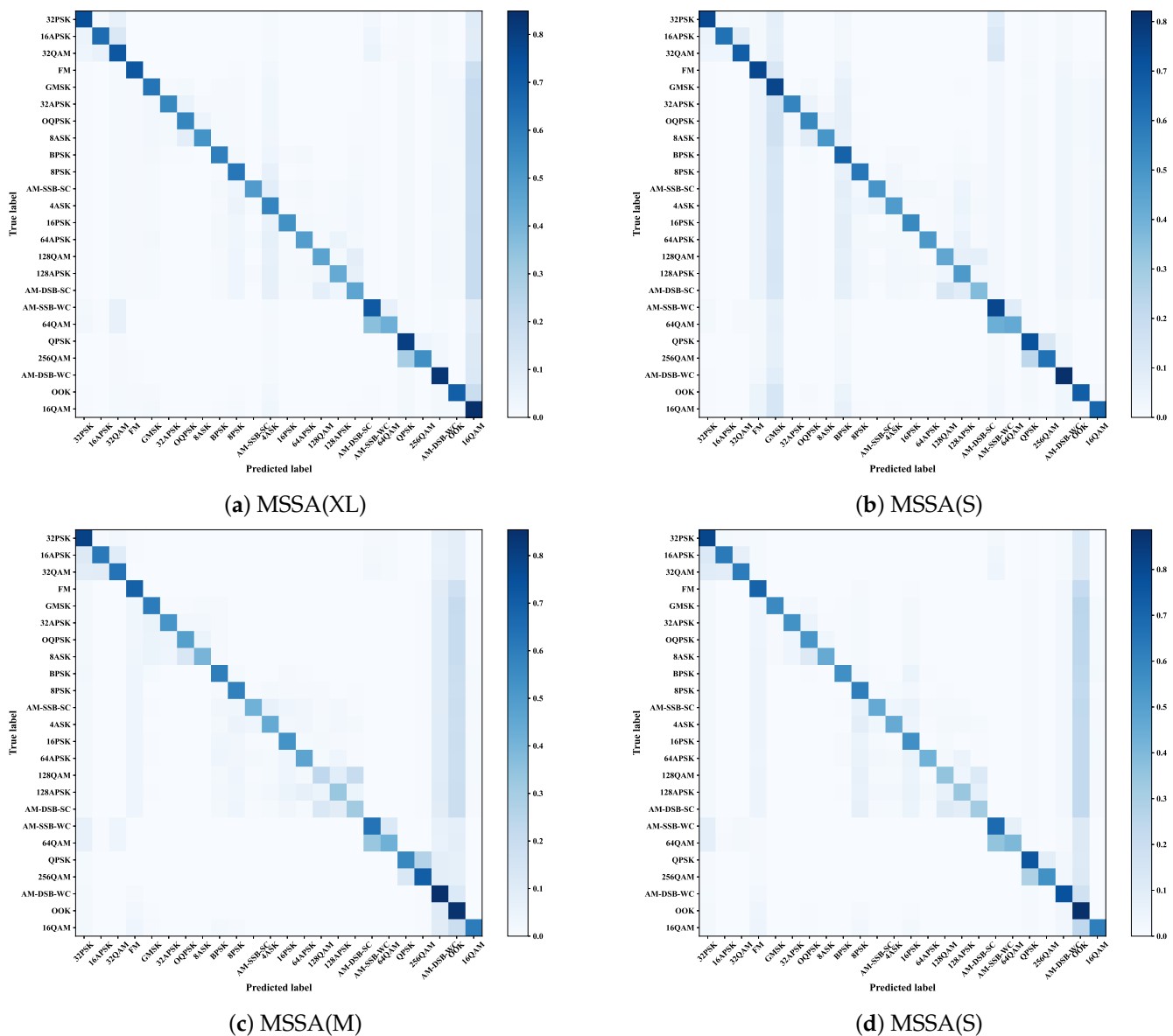

**Figure 11.** Confidence in the confusion matrices of MSSA in different sizes. OOK modulation scheme is still a difficult task for MSSA in M or S size.

## 5. Conclusions

Current AMR methods with large numbers of parameters and high computational complexity are difficult to employ on UAV-to-ground AMR systems. The limited data processing range and low noise resistance also restrict the performances of deep learning methods. Therefore, we proposed MSSA, with fewer parameters, for drone–ground signal processing platforms. We proposed a residual dilated module with a larger receptive field to expand the data processing range and a self-attention module to dynamically acquire information from either local or global contexts, which provides strong noise resistance. Finally, we adjusted the structure of MSSA with different subsampling rates and proposed large, medium, and small MSSA models, which all performed well on the AMR task and had different advantages. The L-sized MSSA(L) was most recommended, with a moderate performance in terms of accuracy, training speed, and the number of parameters. The default MSSA model had the highest accuracy among current models, with a moderately slower training speed than ResNet. Compared with the LSTM, ResNet, and CNN models, MSSA had fewer parameters, making it suitable and scalable for practical applications in drone-based AMR systems with limited computing resources.

As illustrated in Figure 11, the OOK modulation scheme posed the greatest challenge for the M- and S-sized MSSA models, while the MSSA(XL) model had issues recognizing 16QAM modulation. This could possibly be attributed to the subsampling rate of the input samples. Hence, combining data with different subsampling rates may be an effective solution for the AMR task. Future work will explore how to implement this combination and select appropriate subsampling rates.

The signal pattern recognition algorithm proposed in this study demonstrated superior performance in computer simulations and showed promise for deployment on resource-constrained UAV platforms to enable real-time signal analysis. Further verification through physical experiments remains a priority in future research.

**Author Contributions:** Y.S.: conceptualization, methodology, software, writing—original draft preparation, and visualization; H.Y.: conceptualization, resources, writing—original draft, supervision, project administration, and methodology; P.Z.: data curation, and software; Y.L.: validation and resources; M.C.: investigation and formal analysis; J.L.: writing—review and editing. All authors have read and agreed to the published version of the manuscript.

**Funding:** This research received no external funding.

**Data Availability Statement:** Not applicable.

**Conflicts of Interest:** The authors declare no conflict of interest.

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
