# Peer review of "A Multi-Subsampling Self-Attention Network for Unmanned Aerial Vehicle-to-Ground Automatic Modulation Recognition System"

_drones, doi:10.3390/drones7060376_

Round 1

Reviewer 1 Report (Previous Reviewer 3)

The paper has been reviewed again (re-submission) and it would be interesting to be published. 

Reviewer 2 Report (Previous Reviewer 1)

The paper can be accepted as it is.

The paper is well written. The contributions are well explained and positioned in relation to the literature. 

Reviewer 3 Report (Previous Reviewer 2)

The authors have answered all my questions. I recommend to accept the paper. 

This manuscript is a resubmission of an earlier submission. The following is a list of the peer review reports and author responses from that submission.

Round 1

Reviewer 1 Report

This paper investigates deep learning applications of radio automatic modulation recognition (AMR) applications in unmanned aerial vehicle (UAV)-to-ground AMR systems. Comments for improving the manuscript are as follows:

1. The authors propose a multi-subsampling self-attention (MSSA) network for UAV-to-ground AMR systems in Section 3. Could you provide more explanation or insight on your proposed network?

2. It is highly suggested to improve the quality of some figures. These figures can be easily improved for changing their quality, e.g., the eps files can be used in this work.

3. The authors comprehensively portrayed their solution, but did not provide enough details on the state-of-the-art and motivation/rationale on the proposed approach (i.e. specific differences and distinction to the already proposed approaches).

4. Also, the proposed solution is tested with for one specific application scenario, while the other possible scenarios are not mentioned or discussed (if not observed in details). Some explanation on this matter should be given.

5. The system model (i.e., Fig. 1) is well described. However, it misses some concrete examples to help the reader to understand more quickly the approach. Therefore, one or two illustrative examples should be added. In addition, could you provide more explanation or insight on your considered model?

6. In Sec. 4, I am not sure why the parameters were chosen that way. Are they the typical ones chosen for these kinds of scenarios? In general, you should clearly explain the rationale for selecting both the simulation parameters and metrics that you have chosen.

7. I think that its relationship with the topic of “UAV-Assisted Intelligent Vehicular Networks” remains weak. Some recent papers working on UAV-assisted networks should be mentioned and discussed:

a) Computing in the Sky: A Survey on Intelligent Ubiquitous Computing for UAV-Assisted 6G Networks and Industry 4.0/5.0. Drones. 2022; 6(7):177.

b) "A V2I and V2V collaboration framework to support emergency communications in ABS-aided Internet of Vehicles," in IEEE Transactions on Green Communications and Networking, Early Access.

c) "Flying Social Networks: Architecture, Challenges and Open Issues," in IEEE Network, vol. 35, no. 5, pp. 242-248, September/October 2021

The authors should revise and carefully proofread the paper in terms of punctuation and typos. For example, the abbreviation “SNR” is not defined when mentioned in the abstract. The paper should be rechecked for the similar errors.

Reviewer 2 Report

The authors investigated deep learning applications of radio automatic modulation recognition (AMR) applications in unmanned aerial vehicle (UAV)-to-ground AMR systems. A multi-subsampling self-attention (MSSA) network for UAV-to-ground AMR systems is proposed, for which we devise a residual dilated module containing ordinary and dilated convolution to expand the data processing range, followed by a self-attention module to improve classification even in the presence of noise interference. Moreover, the signals are subsampled to reduce the number of parameters and amount of calculation. Three model sizes are proposed, namely large, medium, and small. Simulation results verify the effectiveness of proposed schemes. The paper is in good writing and the technical points are clear and innovative. However, there are still several points to be addressed.

1. what is the difference between multi-subsampling and sub-sampling?

2. How and why to amplify the influence of parameters that are beneficial for modulation recognition?

3. What is the difference between I/Q proposed here from traditional magnitude and phase data of a signal?

4. Basically, the features of signal have not been focused too much. The model does not show how recognition is affected and improved by virtue of these signal features. In other words, it is a general model.

5. The organization and presentation are generally good.    

The overall presentation is good. 

Reviewer 3 Report

The paper presents deep learning applications of radio automatic modulation recognition (AMR) applications in unmanned aerial vehicle (UAV)-to-ground AMR systems. Specifically, the authors propose a multi-subsampling self-attention (MSSA) network for UAV-to-ground AMR systems, for which they devise a residual dilated module containing ordinary and dilated convolution to expand the data processing range, followed by a self-attention module to improve classification even in the presence of noise interference. stance. Finally, they adjusted the structure of MSSA with different subsampling rates and proposed large, medium, and small MSSA models, which all performed well on the AMR task and had different advantages. The L-sized MSSA(L) was most recommended, with moderate performance in terms of accuracy, training speed, and the number of parameters. The default MSSA model had the highest accuracy among current models, with a moderately slower training speed than ResNet. Compared with LSTM, ResNet, and CNN models, MSSA had fewer parameters, making it suitable and scalable for practical application in drone-based AMR systems with limited computing resources.

The paper has a good structure and both theoretic and experimental sections are well presented. The paper also has numerous tables and figures where the research work is analyzed. Finally, the authors present an interesting bibliography.
